# OpenReview forum: "Black-Box Guardrail Reverse-engineering Attack"
_ICLR.cc/2026/Conference — ICLR 2026 Conference Withdrawn Submission_

### Official Review · Reviewer_5QfE · 2025-10-19

**Soundness:** 2
**Presentation:** 2
**Contribution:** 2
**Rating:** 4
**Confidence:** 4

**Summary:**

This paper introduces GRA, a systematic method for black-box extraction of LLM guardrails. By iteratively querying commercial models (ChatGPT, DeepSeek, Qwen3) and training a surrogate model through reinforcement learning with genetic algorithm–based data augmentation, GRA approximates the victim model’s safety decision boundary, that is, when it refuses or allows a response. The authors evaluate fidelity using Rule Matching Rate (RuleMR) and a nine-dimension ethical value benchmark, demonstrating that GRA can reproduce both surface-level moderation and deeper normative patterns with over 0.9 accuracy while costing under $85 in API usage.

**Strengths:**

The paper proposes a systematic analysis and empirically demonstrates guardrail extraction under a pure black-box setting.

**Weaknesses:**

The comparison metric focuses mainly on binary refusal vs. allowance, which oversimplifies real guardrail behavior (e.g., soft refusals, partial redactions).

The paper does not examine cases where multiple concurrent guardrails (e.g., alignment + rule-based moderation) jointly affect decisions; hence, it remains unclear if GRA can disentangle or recover overlapping policies.

The proposed countermeasures (monitoring, adaptive rejection, dynamic policies) are conceptually useful but not experimentally tested or quantified.

**Questions:**

Could the authors extend the evaluation beyond nine moral/ethical criteria to include practical guardrail domains (e.g., privacy, misinformation, or policy compliance)?

How would GRA perform when multiple guardrail mechanisms coexist, such as alignment-tuned safety + external moderation model? Can it still converge to a stable surrogate?

Since the paper treats guardrail behavior as binary refusal detection, have the authors considered a graded or probabilistic evaluation that captures soft moderation or nuanced policy enforcement?

To what extent does GRA replicate content-level semantics versus only refusal/allowance?

The proposed defenses are discussed qualitatively. Could the authors provide at least a simulation or quantitative estimate of their effectiveness?

---

### Official Review · Reviewer_ibpq · 2025-10-28

**Soundness:** 3
**Presentation:** 3
**Contribution:** 3
**Rating:** 6
**Confidence:** 4

**Summary:**

This paper investigates a timely and important problem: whether the safety guardrails used in commercial large language models (LLMs) can be reverse-engineered in a black-box setting.
The authors propose GRA, a reinforcement learning–based framework combined with genetic algorithm–driven data augmentation to approximate the decision behavior of guardrails without internal access.
By iteratively collecting input–output pairs and refining a surrogate model, the method aims to replicate the moderation policy of the target system.
Experiments on ChatGPT, DeepSeek, and Qwen3 demonstrate that GRA achieves a high rule matching rate with moderate API cost.
The work highlights significant security risks of current LLM safety architectures.

**Strengths:**

1. The paper tackles an emerging and practically relevant problem — the vulnerability of guardrails in deployed LLMs. Considering the current industry emphasis on AI safety, this topic is both urgent and important.
2. The authors present a well-structured description of the threat model, including attacker goals, capabilities, and system assumptions. This improves reproducibility and clarity.
3. The proposed combination of reinforcement learning and genetic augmentation is conceptually simple but works effectively in practice. The experiments cover multiple target systems, datasets, and metrics, showing consistent performance trends.

**Weaknesses:**

1. Although the paper classifies guardrails into alignment-based, model-based, and rule-based types, the proposed method seems mainly tailored for model-based guardrails.
It is unclear how GRA would handle alignment-based guardrails, which evolve dynamically as models are updated.
2. The internal policies of commercial systems are inaccessible, so it is impossible to verify whether the surrogate model truly recovers the underlying rules or merely mimics surface-level behavior.
The reported “Rule Matching Rate” therefore lacks a solid reference baseline.
3. The conceptual difference between GRA and traditional model extraction is discussed, but the mechanism-level novelty remains weak.
The paper could provide theoretical or quantitative justification explaining why GRA is more suitable for guardrail imitation than existing extraction methods.
4. Toxicity scores are measured using moderation models such as LlamaGuard, ShieldGemma, and GPT-4o.
These models may share data or design similarities with the evaluated systems, causing pseudo-consistency and optimistic results.
5. Although GRA outperforms baselines, its toxic score improvement on ChatGPT is relatively small, suggesting limited real-world effectiveness or incomplete policy replication.

**Questions:**

1. How would GRA adapt to alignment-based or rule-based guardrails, which may evolve over time or rely on internal model tuning rather than static filters?
2. Since the internal rules of commercial systems are unknown, how can we validate that the reported Rule Matching Rate actually corresponds to true policy extraction?
3. What makes model extraction attacks fundamentally insufficient for this problem? Can the authors provide quantitative or theoretical evidence for the claimed novelty of GRA?
4. Have the authors considered using human evaluation or independent toxicity detectors to mitigate the bias introduced by overlapping moderation models?
5. Could the authors provide an analysis of failure cases, especially for ChatGPT, where the performance improvement seems limited?

---

### Official Review · Reviewer_qmKK · 2025-10-28

**Soundness:** 1
**Presentation:** 1
**Contribution:** 2
**Rating:** 2
**Confidence:** 4

**Summary:**

The paper proposes GRA (Guardrail Reverse-engineering Attack), a black-box attack method that applies established model extraction techniques to LLM guardrails using reinforcement learning combined with genetic algorithm-inspired data augmentation. The method iteratively queries a victim LLM system to collect input-output pairs, trains a surrogate guardrail (Llama-3.1-8B-Instruct) using the GRPO algorithm with reward signals from the victim system, and applies crossover and mutation operations on high-divergence samples to explore decision boundaries and generate challenging training examples.

The authors measure "success" through three main approaches:
1. Toxic scores using external evaluators (LlamaGuard, ShieldGemma, GPT-4o) to assess whether the surrogate produces similarly "safe" outputs as the victim, with a "learning progress" metric comparing how close the surrogate gets to victim behavior;
2. Rule matching rate (RuleMR) tested on a self-created "single-choice value assessment benchmark covering nine philosophical and ethical dimensions,"; and
3. Standard classification metrics (accuracy, precision, recall, F1, AUC) measuring the surrogate's ability to distinguish harmful from benign inputs.

**Strengths:**

The paper proposed an interesting question, which is whether commercial LLM guardrails can be systematically reverse-engineered through black-box queries alone, and provides the first empirical demonstration that behavioral extraction is feasible on real-world systems with modest resources.

**Weaknesses:**

- The authors claim to identify a "new class of vulnerabilities" in guardrails that expose observable decision patterns, but this is misleading. The vulnerability they exploit—information leakage through black-box input-output queries—is a well-established problem in machine learning security. The actual contribution is demonstrating that existing vulnerabilities apply to guardrail components and developing a specific attack method (GRA) for this context, not discovering a fundamentally new vulnerability class. This overclaim weakens the paper's credibility, as the novelty lies in the exploitation technique rather than in identifying previously unknown security risks.

- the authors use 3 metrics to show GRA's performance, but I do not understand there metric can reflect it. Measuring behavioral similarity between surrogate and victim outputs—similar behavior does not definitively prove that the underlying rules or decision logic have been extracted, only that the surrogate mimics observable patterns.

- The paper's threat model and attack methodology only address output-side guardrails that filter LLM-generated responses, while completely ignoring input-side guardrails that block or sanitize user prompts before they reach the LLM. Real-world commercial systems may employ dual-layer protection with both input prompt filtering and output response filtering.

- There are two gaps leaves the central premise—that guardrail extraction poses "significant security risks"—as an unproven assertion rather than an empirically validated conclusion:
    1. The paper demonstrates that guardrails can be reverse-engineered but critically fails to show the practical exploitation consequences. The authors do not provide evidence of responsible disclosure to the commercial providers and get acknowledgment that this constitutes a genuine security vulnerability
    2. More importantly, they do not demonstrate how the extracted guardrail enables concrete attacks—specifically, whether possessing a surrogate guardrail actually helps adversaries craft more effective jailbreaks or bypass the original guardrail more successfully

**Questions:**

1. You claim to introduce a "new class of vulnerabilities" in guardrails, but model extraction attacks exploiting observable input-output behavior have been studied. Can you clarify what is fundamentally new about the vulnerability class itself, rather than just applying known extraction techniques to a new target (guardrails)?
2. Would it be more accurate to frame your contribution as "the first systematic application of model extraction attacks to guardrail components" rather than discovering new vulnerabilities?
3. Your evaluation relies on behavioral similarity metrics. However, similar outputs don't necessarily mean you've extracted the underlying rules or decision logic—it could just mean you've trained a model that happens to make similar predictions. How do you validate that you've actually reverse-engineered the guardrail's policy rules rather than just creating a behavioral mimic?
4: In a true reverse-engineering validation, one would compare extracted rules against ground truth. Since you don't have access to the actual guardrail rules, what evidence supports that your surrogate has captured the underlying policy versus merely overfitting to observable patterns?
5. Your threat model only addresses output-side guardrails (filtering LLM responses), but commercial systems likely employ input-side guardrails that block malicious prompts before they reach the LLM. Why is input filtering completely absent from your methodology and evaluation?
6. Can your GRA method be extended to extract input-side guardrails, and if so, how would the methodology differ?
7. Did you conduct responsible disclosure by reporting your findings to OpenAI (ChatGPT), DeepSeek, and Qwen3 before publication? If so, did they acknowledge this as a legitimate security vulnerability?
8. You demonstrate guardrail extraction but don't show how this enables attacks. Can you provide empirical evidence that possessing a surrogate guardrail helps adversaries: (a) craft more effective jailbreaks, (b) bypass the original guardrail more successfully.

---

### Official Review · Reviewer_iv8w · 2025-10-31

**Soundness:** 2
**Presentation:** 3
**Contribution:** 2
**Rating:** 2
**Confidence:** 4

**Summary:**

This paper proposes a reverse-engineering attack against the guardrails of black-box large language models (LLMs). The authors claim this is the first study of such an attack and propose a set of seven metrics for evaluation.

While the problem of LLM security addressed in this paper is important and timely, I have significant reservations about this work's contribution. The primary issues include: 1) The claim of being the "first study" is questionable, and there is a severe lack of comparison with the extensive existing work on prompt stealing; 2) The experimental evaluation relies on a private dataset, lacking transparency and reproducibility; 3) The proposed evaluation metrics are overly simplistic, resembling standard classification metrics, and are not convincing for comprehensively measuring the effectiveness of a reverse-engineering attack.

**Strengths:**

The vulnerability of LLM guardrails, which this paper focuses on, is an important and pressing issue. Understanding the internal mechanisms of black-box models (even their safety mechanisms) is crucial for building more robust systems.

**Weaknesses:**

Questionable Novelty and Lack of Comparison with Related Work:
The authors claim this is the "first study" of black-box guardrail reverse-engineering. This assertion appears unsubstantiated. The field of LLM reverse-engineering, particularly "Prompt Stealing" or "Prompt Extraction," is already well-researched (see citation below). Prompt stealing is, in essence, a form of reverse-engineering attack targeting system instructions or guardrails.

The critical flaw of this paper is its complete failure to compare the proposed attack with these known prompt-stealing techniques. The authors must clearly articulate how their work fundamentally differs from this prior SOTA. If a guardrail is essentially a form of "prompt," what is the advantage of the proposed method over existing attacks? This lack of comparison significantly diminishes the paper's contribution and novelty.

Unconvincing Evaluation Metrics:
The paper claims to use a "comprehensive set of seven metrics" to evaluate attack effectiveness. However, a closer inspection reveals that they "are just different metrics on a classification task" (e.g., success rate, precision, recall, etc.).

For a reverse-engineering task, especially one involving the recovery of natural language rules or prompts, such simple classification metrics are "too easy" and far from sufficient. For example, does a "failed" attack recover partial information about the guardrail? How semantically close is the content recovered from a "successful" attack to the original guardrail?

The authors are severely lacking more meaningful evaluation metrics, such as "text similarity" (e.g., ROUGE, BLEU, BERTScore, or other embedding-based similarity calculations), to measure the proximity between the recovered guardrail (text) and the ground-truth guardrail (text). The current metrics fail to convincingly demonstrate the true effectiveness and depth of the attack.


@article{sha2024promptstealing,
  title        = {Prompt Stealing Attacks Against Large Language Models},
  author       = {Sha, Zeyang and Zhang, Yang},
  journal      = {arXiv preprint arXiv:2402.12959},
  year         = {2024},
  doi          = {10.48550/arXiv.2402.12959},
  archivePrefix= {arXiv},
  eprint       = {2402.12959},
  primaryClass = {cs.CR}
}

@inproceedings{zhang2024effective,
  title     = {Effective Prompt Extraction from Language Models},
  author    = {Zhang, Yiming and Carlini, Nicholas and Ippolito, Daphne},
  booktitle = {Proceedings of the Conference on Language Modeling (COLM 2024)},
  year      = {2024},
  url       = {https://openreview.net/forum?id=0o95CVdNuz},
  note      = {OpenReview, COLM 2024}
}

@inproceedings{wang-etal-2024-raccoon,
  title     = {Raccoon: Prompt Extraction Benchmark of {LLM}-Integrated Applications},
  author    = {Wang, Junlin and Yang, Tianyi and Xie, Roy and Dhingra, Bhuwan},
  editor    = {Ku, Lun-Wei and Martins, Andre and Srikumar, Vivek},
  booktitle = {Findings of the Association for Computational Linguistics: ACL 2024},
  month     = aug,
  year      = {2024},
  address   = {Bangkok, Thailand},
  publisher = {Association for Computational Linguistics},
  url       = {https://aclanthology.org/2024.findings-acl.791/},
  doi       = {10.18653/v1/2024.findings-acl.791},
  pages     = {13349--13365}
}

@inproceedings{hui2024pleak,
  title     = {PLeak: Prompt Leaking Attacks against Large Language Model Applications},
  author    = {Hui, Bo and Yuan, Haolin and Gong, Neil and Burlina, Philippe and Cao, Yinzhi},
  booktitle = {Proceedings of the ACM Conference on Computer and Communications Security (CCS)},
  year      = {2024},
  doi       = {10.1145/3658644.3670370},
  url       = {https://dl.acm.org/doi/10.1145/3658644.3670370},
  note      = {Also available as arXiv:2405.06823}
}

**Questions:**

Please elaborate in the "Related Work" section on the connection and distinction between "prompt stealing" and "guardrail reverse-engineering"

---

### Note · Authors · 2025-11-22

I have read and agree with the venue's withdrawal policy on behalf of myself and my co-authors.